# Sour Beer as Bioreservoir of Novel Craft Ale Yeast Cultures

**DOI:** 10.3390/microorganisms11092138

**Published:** 2023-08-23

**Authors:** Chiara Nasuti, Jennifer Ruffini, Laura Sola, Mario Di Bacco, Stefano Raimondi, Francesco Candeliere, Lisa Solieri

**Affiliations:** 1Department of Life Sciences, University of Modena and Reggio Emilia, Via Amendola, 2-Pad. Besta, 42122 Reggio Emilia, Italy; chiara.nasuti@unimore.it (C.N.); 227818@studenti.unimore.it (J.R.); 2Department of Life Sciences, University of Modena and Reggio Emilia, Via Campi 51, 41125 Modena, Italy; laura.sola@unimore.it (L.S.); stefano.raimondi@unimore.it (S.R.); francesco.candeliere@unimore.it (F.C.); 3Ca’ Del Brado Brewery, Via Andrea Costa, 146/2, 40065 Rastignano, Italy; quark@cadelbrado.it; 4National Biodiversity Future Center (NBFC), 90133 Palermo, Italy

**Keywords:** sour beer, craft brewing, STA1 gene, dextrin, hybrid, *Saccharomyces cerevisiae*, *Saccharomyces uvarum*, *Pichia membranifaciens*

## Abstract

The increasing demand for craft beer is driving the search for novel ale yeast cultures from brewing-related wild environments. The focus of bioprospecting for craft cultures is to identify feral yeasts suitable to imprint unique sensorial attributes onto the final product. Here, we integrated phylogenetic, genotypic, genetic, and metabolomic techniques to demonstrate that sour beer during aging in wooden barrels is a source of suitable craft ale yeast candidates. In contrast to the traditional lambic beer maturation phase, during the aging of sour-matured production-style beer, different biotypes of *Saccharomyces cerevisiae* dominated the cultivable in-house mycobiota, which were followed by *Pichia membranifaciens*, *Brettanomyces bruxellensis*, and *Brettanomyces anomalus*. In addition, three putative *S. cerevisiae* × *Saccharomyces uvarum* hybrids were identified. *S. cerevisiae* feral strains sporulated, produced viable monosporic progenies, and had the *STA1* gene downstream as a full-length promoter. During hopped wort fermentation, four *S. cerevisiae* strains and the *S. cerevisiae* × *S. uvarum* hybrid WY213 exceeded non-*Saccharomyces* strains in fermentative rate and ethanol production except for *P. membranifaciens* WY122. This strain consumed maltose after a long lag phase, in contrast to the phenotypic profile described for the species. According to the *STA1*+ genotype, *S. cerevisiae* partially consumed dextrin. Among the volatile organic compounds (VOCs) produced by *S. cerevisiae* and the *S. cerevisiae* × *S. uvarum* hybrid, phenylethyl alcohol, which has a fruit-like aroma, was the most prevalent. In conclusion, the strains characterized here have relevant brewing properties and are exploitable as indigenous craft beer starters.

## 1. Introduction

Craft beer production has gained increasing interest in many countries as an alternative to mainstream beer [1]. Craft brewing refers to relatively small breweries that aim to produce beers with unique and original sensory characteristics based on quality and diversity [2]. Special flavors increase perceived quality compared to commercial beer [3,4].

The sensory profile of beer is determined by the choice of malt or adjunct, hop variety, water quality and yeast, which catalyzes metabolic reactions during wort fermentation, giving the beer its flavor profile. Yeasts are responsible not only for sugar consumption and ethanol and CO_2_ production but also for releasing aroma-active secondary side-products of alcoholic fermentation—higher alcohols and ethyl and acetate esters—that contribute to solvent-like and fruity notes and a sweet taste [5]. They also can transform neutral precursors in the wort and hops—glycosides, polyfunctional thiols, and terpenes—into sensorial active compounds, as reviewed by Svedlund et al. [6]. This bio-flavoring aptitude qualitatively and quantitatively varies depending on the strain and species employed [7]. However, the pool of commercial *Saccharomyces cerevisiae* for ale beer and *Saccharomyces pastorianus* for lager beer is restricted and genetically redundant [8], hindering the chance to diversify a beer’s sensory profile through yeast bio-flavoring.

To expand the number of craft brewing starters and to diversify the portfolio of beer styles, various GMO-free strategies have been proposed. These approaches include hybridization, use of non-*S. cerevisiae* strains, and bioprospecting for novel wild yeast species and strains from alternative or unusual niches [9,10,11]. Recently, genetically modified yeasts have been exploited to boost hop-related flavor-active molecules [12,13,14,15]. The selection of suitable yeast strains is a crucial step in producing high-quality craft beer that affects both wort fermentation and beer conditioning. Yeast strains significantly affect the production of volatile organic compounds (VOCs), which are essential for a beer’s flavor and aroma [2,3], particularly for craft beers, which are often unfiltered, unpasteurized, and bottle-conditioned [4,5].

In craft brewing, interest in the sour beer style has been growing [16]. Traditionally fermented sour beers, such as Belgian-type lambics, pass through spontaneous fermentation and wood maturation, which can take up to three years. This process involves conventional and unconventional yeasts, lactic acid bacteria, and, to a lesser extent, acetic acid bacteria [16,17,18,19,20]. These microorganisms are spontaneous beer contaminants arising from raw materials, air, and the brewery environment and equipment, such as the inner surface of barrels [21]. Recently, they have been revised as “drivers” of innovation and diversification of the sensory profile in sour beers [22]. Other craft brewing strategies mimic lambic-style beer in complexity, fruitiness, and sourness, but they reduce maturation time and increase fermentation control, as reviewed by [16,23]. Among these methods, ale beer can be soured by a primary fermentation with brewery yeast pure cultures, followed by maturation aging in wooden barrels, where the implantation of a complex house microbiota can lead to acidic and peculiar sensory profiles, the so-called sour-matured production styles.

In this study, we considered the in-house microbiota of sour ale beers during aging as a source for novel brewing cultures. For this purpose, we taxonomically identified and technologically profiled the cultivable microbial fraction present in sour ale beer during maturation in wooden barrels and identified *Saccharomyces* wild yeast candidates to be exploited as starters in wild ale beer production.

## 2. Materials and Methods

### 2.1. Chemicals, Media, and Reference Strains

Chemicals were purchased from Sigma Aldrich (St. Louis, MO, USA), unless otherwise stated. Media and anaerobic systems were purchased from Oxoid (Cambridge, UK), while reagents for molecular biology from Thermo Fisher Scientific (Waltham, Massachusetts, United States). Oligonucleotide synthesis and Sanger sequencing were carried out by Bio-Fab Research core facility (Rome, Italy).

Reference strains are listed in Table 1.

### 2.2. Sour Beer Production and Sampling

The sour beer was produced at a craft brewery in Bologna, Italy, and Figure 1 illustrates the semi-spontaneous fermentation method employed.

Three distinct beer products labeled as 1, 2, and 3 were blended in equal proportions (1:1:1). Specifically, beer 1 was produced by the spontaneous fermentation of a wort (cereal grist: barley malt 72%, unmalted wheat 15%, rye 8%, cara malt 5%) with an adjunct of fresh whole grapes with skins (grapes/wort ratio 1:10) and an aging of 8 months in barrel. Beer 2 was produced through a wort fermentation (same grist as beer 1) with the single strain *Brettanomyces bruxellensis* WLP650 (White Lab), a subsequent aging of 5 months in barrels, and a final maceration (2 months) of fresh whole cherries. Beer 3 was produced with a cereal grist of barley malt 62%, unmalted wheat 30%, and oats 8% that underwent primary fermentation with a blend of three commercial *S. cerevisiae* strains (WLP565, Atecrem, and 3724; Table 1), followed by barrel maturation for 10 months. A blended sample was shipped to the laboratory under refrigerated conditions and immediately submitted for microbiological and physicochemical determinations.

### 2.3. Microbial Counts, Bacterial and Yeast Isolation

After a ten-fold dilution in saline (9 g/L NaCl), appropriate sample dilutions were spread on Petri dishes. Yeast counts and isolation were performed on YPDA (yeast extract 1% *w/v*, dextrose 2% *w/v*, agar 2% *w/v*) and Wallerstein Laboratory (WL) media supplemented with chloramphenicol (100 μg/mL) and then incubated for 24–48 h at 28 °C. Man-Rogosa–Sharpe medium supplemented with maltose (5 g/L) and cycloheximide (20 mg/mL) (mMRS) was used for lactic acid bacteria (LAB) isolation. The anaerobic incubation was set for 48–72 h at 30 °C.

Yeast and LAB populations were quantified in triplicate by counting plate colonies ranging from 20 to 200 and expressing the results as the mean log_10_ CFU/mL. Colonies were purified through at least double streaking on the same isolation medium. Cultures were microscopically examined for cellular morphology, and bacterial isolates were further screened for Gram staining and catalase activity. For long-term preservation, isolates were cryopreserved at −80 °C using the corresponding isolation medium supplemented with 25% (*v/v*) glycerol.

### 2.4. Yeast and LAB identification

Yeast genomic DNA was extracted according to Hoffman and Winston [24], while for LAB DNA it was according to Gala et al. [25]. The integrity of DNA was checked by electrophoresis on a 0.8% (*w/v*) agarose gel in 0.5X TBE buffer (45 mM Tris–HCl, 45 mM boric acid, and 1 mM EDTA, pH 8.0), and DNA quality and quantity were evaluated spectrophotometrically using a NanoDrop ND-132 1000 device (Thermo Fisher Scientific, Waltham, MA, USA). All DNA samples were diluted to a concentration of 50 ng/μL with ddH_2_O for further analysis.

The primers, reaction mixtures, and cycling conditions of PCR amplification reactions are detailed in Appendix A. Concerning yeast identification, the region including the 5.8S rRNA gene and the upstream and downstream internal transcribed spacers (globally referred to as ITS region) was PCR amplified in a final volume of 40 μL using the primers ITS1 and ITS4, as described by White et al. [26]. The restriction fragment length polymorphism (RFLP) analysis of ITS amplicons was carried out with the endonucleases *Hae*III and *Hinf*I [27] and the length of PCR amplicons and RFLP fragments were compared to those in the YeastID database (www.yeast-id.org; last accessed on 14 December 2022) using rank parameters at ±10. Discrimination of *S. cerevisiae*, *S. eubayanu*s, and *S. uvarum* species was carried out according to the species-specific PCR method previously described [28,29]. All primer concentrations were set to 0.4 μM, except for SbayR1, which was used at a final concentration of 0.8 μM. Strains representative of each ITS RFLP and species-specific PCR profile were subjected to PCR amplification of the D1/D2 region of the 26S rRNA (LSU) gene according to Martini et al. [30].

Concerning LAB identification, 16S rRNA gene was PCR amplified with the primers 27f and 1490r [31]. Amplified 16S Ribosomal DNA Restriction Analysis (16S-ARDRA) was performed by the endonucleases *Hha*I, *Hinf*I, and *Tru1*I, and the resulting 16S-ARDRA profiles were compared to the in silico profiles obtained for the most common LAB beer species (Appendix A).

### 2.5. Phylogenetic Analysis

Phylogenetic analyses of yeast and bacteria strains were performed using D1/D2 LSU and 16S rRNA sequences, respectively. Amplicons were purified using DNA Clean & Concentrator™-5 Kit (Zymo Research, Orange, CA, USA) and sequenced using both amplification primers. Contig sequences were merged using the program SeqMan (DNASTAR, Madison, WI, USA) and poor-quality ends were edited manually to remove primers. The resulting contig sequences were used as queries in a BLASTn search against the NCBI RefSeq database [32] (last accessed on 14 February 2023). Cut-offs of 98.7% and 99% gene similarity were used for bacterial and yeast species attribution, respectively [33,34]. The sequences were aligned with Muscle program [35] in MEGA X software [36] and the resulting alignments were subjected to a DNA substitution model analysis to select the best-fitting model. Phylogenetic relationships were inferred using the Neighbor joining method [37]. Confidences for the phylogenetic tree were estimated from bootstrap analysis (1000 replicates). Trees were visualized using Interactive Tree of Life (ITOL) [38] and rooted at outgroup reference species. The sequence data of D1/D2 LSU regions were deposited in the GenBank NCBI database under accession numbers OM618137 to OM618146, while the accession numbers for the 16S rRNA gene sequences were OM618118 to OM618120.

### 2.6. Genotyping Analysis

All the LAB and yeasts isolates were genotyped with a repetitive extragenic palindromic–polymerase chain reaction (rep-PCR) fingerprinting analysis based on the (GTG)_5_ primer (5′-GTG GTG GTG GTG GTG-3′) [39], as previously described [40,41].

Commercial ale beer starters used in primary fermentation (Figure 1) and listed in Table 1 were genotyped by three different methods: (GTG)_5_ rep-PCR, inter-delta PCR, and R3-RAPD. (GTG)_5_ rep-PCR was carried out as reported above. PCR amplification of LTR regions near the TY1 and TY2 retrotransposon (referred to as inter-delta regions) was carried out with primers d12 (5′-TCAACAATGGAATCCCAAC-3′) and d21 (5′-CATCTTAACACCGTATATGA-3′) [42], while R3-RAPD PCR amplification was carried out using the primer R3 (5′-ATG-CAGCCAC-3′) [43] (Appendix A).

(GTG)_5_ rep-PCR profiles were analyzed by electrophoresis on a 1.8% (*w/v*) agarose gel (25 × 27 cm) in 0.5X TBE buffer at a constant 70 V for 6 h under cold conditions; inter-delta and R3-RAPD profiles on a 1.5% (*w/v*) agarose gel (25 × 15 cm) in 0.5X TBE buffer at a constant 70 V for 150 min under cold conditions. In all genotyping tests, agarose gels were stained with SYBR™ Safe DNA Gel Stain (Invitrogen, Waltman, MA, USA) and GeneRuler 100 bp Plus DNA Ladder™ was used as a molecular weight standard. After visualization under UV light, digitalized images were analyzed using the BioNumerics software v8.10 (Applied Maths, Sint-Martens-Latem, Belgium). Pearson’s correlation similarity coefficient was used to convert computed band patterns into similarity matrix. Optimization and curve smoothening parameters were optimized with the specific script present in BioNumerics. The unweighted-pair group method analysis using the arithmetic means (UPGMA) was used for tree construction. Yeasts and LAB isolates with ≥92 and 85% similarity values, respectively, were treated as a single strain.

### 2.7. Flocculation, Sporulation Assays and MAT Locus Genotyping

A flocculation assay was performed as reported in [44]. Briefly, yeasts were propagated in YPDA medium until OD_600nm_ > 2.0. Three mL of pure liquid cultures were centrifuged at 2000× *g* for 5 min, and cells were resuspended in an equal volume of flocculation buffer (50 mmol/L Na acetate/acetic acid, 5 mmol/L CaSO_4_; pH 4.5). The OD_600nm_ values were spectrophotometrically measured immediately after resuspension, as well as after 15 and 30 min of incubation at room temperature without agitation. To assess flocculation ability, the following equation was used:F (%) = (OD_600nm_ after 15 or 30 min/OD6_00nm_ at starting time) × 100,(1)

Yeasts were classified according to F% as follows: score from 0 to 10%, “very flocculant”; from 10 to 30%, “moderately flocculant”; from 70 to 90%, “poorly flocculant”; from 90 to 100%, “non-flocculant”.

To induce sporulation, exponentially grown cells were sub-cultured to acetate agar plates (2% *w/v* agar, 0.6% *w/v* sodium acetate) [45] and incubated at 28 °C for two weeks. Asci formation was checked and counted with a Bürker chamber after 3 and 14 days using an optical microscope with 400× magnification. Sporulation percentage was calculated as follows:Sporulation (%) = [(n° asci)/(n° asci + n° cells)] × 100,(2)

Spore viability was determined by asci dissection and monospore clone constitution as previously reported [46], according to the following equation:Spore Viability % = (n° of vital monosporic cultures/n° of dissected spores) × 100,(3)

For each strain, at least eight asci were dissected. Both wild strains and their monospore derivatives were genotyped for the *MAT* locus according to Catallo et al. [46]. Strains BY4742 and BY4743 (Table 1) were used as positive *MATα* and *MATα*/*MAT*a controls, respectively.

### 2.8. Maltose Fermentation Tests

Preliminary, maltose fermentation test was carried out according to Kurtzman et al. [45]. A sterile basal medium (0.45% *w/v* yeast extract, 0.75% *w/v* peptone and 0.0048% *w/v* bromothymol blue as pH indicator) (4.5 mL) was supplemented with a filter-sterilized stock sugar solution (either maltose or glucose) at a final concentration of 2% (*w/v*) and dispensed in test tubes with screw caps with an upside-down inverted Durham tube inserted. Each yeast strain was propagated in a YPD medium at 27 °C for 24 h and inoculated at the final concentration of 2 × 10^7^ CFU/mL. Tubes were incubated at 28 °C, and gas production was checked after 3, 7, and 14 days. Scores were attributed as follows: +, strongly positive, insert filled within 7 days; s, slowly positive, insert slowly filled after more than 7 days; −, no gas production. Cells inoculated in basal medium with 2% (*w/v*) glucose were used as a control.

Maltose consumption tests were carried out in triplicate in 100 mL Erlenmeyer flasks containing 45 mL of a Yeast Nitrogen Base (YNB) (0.67% *w/v*) medium supplemented with 2% (*w/v*) of either glucose or maltose and sealed with cotton plug, as previously reported [46,47]. After incubation for 48 h at 25 °C under shaking conditions (150 rpm), optical density (OD) values at 600 nm (OD_600nm_) were measured using a UV-Vis photometer.

### 2.9. Wort Preparation and Micro-Scale Fermentation Trials

For the wort preparation, Pilsen malt (195 kg) (Bestmalz, Heidelberg, Germany), unmalted wheat (130 kg), unmalted oat (*Avena sativa*) (32 kg), and hop pellets Hallertau Perle (Hopsteiner, Mainburg, Germany) were mixed in a final volume of 1600 L. The “liquor-to-grist” ratio was maintained at 3:1. The mash temperature steps involved 10 min at 45 °C, 20 min at 55 °C, 40 min at 66 °C, and 10 min at 78 °C. The boiling process was carried out for 60 min. The resulting hopped wort (11.5 °Plato original gravity; density 1.07 g/L, pH value of 5.15 ± 0.02) was pasteurized at 100 °C for 10 min, and subsequently frozen at −20 °C until use.

Microscale wort fermentation trials were conducted in duplicate using 200 mL glassware bottles equipped with airlocks and filled with 100 mL of pasteurized hopped wort. A single colony of each yeast candidate was propagated at 27 °C for 48 h in YPD medium under orbital shaking (150 rpm). After recovery by centrifugation for 6 min at 2500× *g*, cells were resuspended in pasteurized wort and inoculated in wort at the final concentration of 10^7^ cells/mL. Fermentation bottles were maintained at 20 °C for 21 days and monitored daily by CO_2_ release, weighting the fermenters to track weight loss. The un-inoculated beer wort was used as a negative control.

The fermentation curves were modeled based on the CO_2_ evolution using the “grofit” package for R [48]. The maximum rate of fermentation (µ), maximum fermentation efficiency (the total amount of CO_2_ released at the end of the fermentation (A)), and the time of the lag phase (λ) were determined using parametric fitting methods in “grofit”.

### 2.10. Physicochemical Analysis

The pH was determined at a sample temperature of 25 °C (Crison Instruments, Barcelona, Spain).

Carbohydrates and organic acids in sour beer, wort, and fermented broths, clarified by centrifugation (13,000 rpm for 5 min at 4 °C) and filtration at 0.22 µm, were analyzed in an HPLC system equipped with a refractive index detector (1200 System, Agilent Technologies, Waldbronn, Germany). Elution in isocratic condition was carried out at 60 °C with 0.6 mL/min of 5 mM H_2_SO_4_ through an ion exclusion column (Aminex HPX-87 H, Bio-Rad, Hercules, CA, USA).

Volatile organic compounds (VOCs) were traced by solid-phase microextraction (SPME) followed by gas chromatography-mass spectrometry analysis (GC–MS). Two mL of samples were spiked with 100 µL of 1-chloro-2-fluoro-benzene 5 mg/L, as the internal standard, and the headspace was exposed to a divinylbenzene/carboxen/polydimethylsiloxane fiber (DVB/CAR/PDMS Supelco; Sigma-Aldrich, St. Louis, MO, United States) for 1 h at 60 °C. Analytes were desorbed in a GC–MS apparatus (7820–5975; Agilent Technologies, Santa Clara, CA, United States) equipped with a DB-624 column (30 m × 250 mm × 1.4 mm, Agilent Technologies). Separation was achieved with 1.3 mL/min helium flux following a thermal gradient: 2 min isotherm at 50 °C; 6 °C/min increase to 110 °C; 10 °C/min increase to 240 °C; and 4 min isotherm at 240 °C. NIST 14 spectral library was used to infer the identity of the compounds by comparing their mass spectra. The peak areas of compounds, normalized by the area of the inner standard, were taken to be proportional to their abundances.

### 2.11. Screening of STA1 Gene

Screening for the *STA1* gene was conducted by endpoint PCR using the SD-5A (5-CAACTACGACTTCTGTCATA-3) and SD-6B (5-GATGGTGACGCAATCACGA-3) primers [49], while the analysis of the promoter region, particularly the *UAS2* portion, was conducted with the primers STA1_UAS_Fw (5′-CCTGGCTCAAATTAAACTTTCG-3′) and STA1_UAS_Rv (5′-ACCACCAATAGGCAATAGAAA-3′) [50]. PCR mixtures and cycling conditions are reported in Appendix A. PCR amplicon purification, sequencing and sequence analysis were carried out as reported above.

### 2.12. Statistical Analysis

The datasets were analyzed using GraphPad Prism 8 software (GraphPad Software Inc., San Diego, CA, USA) through one-way or two-way ANOVA. A *p*-value threshold of 0.05 was set for both analyses. All values presented in this study are the means of three replicates ± standard deviation (SD).

Principal component analysis (PCA) was applied to examine the data matrix of VOC profiles, presenting a size {20, 53}, consisting of the 53 relative areas of VOCs, determined for the two independent fermentations with the 9 yeast strains in addition to the initial wort. A screen plot was used to establish the number of significant principal components (PCs) based on the percentage of variance explained. Scores and loadings of the PCA model were calculated using Past version 4.12 [51], as well as UPGMA hierarchical clustering of samples based on the Bray–Curtis similarity index of VOC profiles.

## 3. Results

### 3.1. Sour Beer Physicochemical Caracterization and Microbiological Analysis

The samples were collected during the late maturation phase. Sour beers are distinguished from ales and lagers by their elevated levels of organic acids, specifically lactic and acetic acid. These contribute to a lower pH and a more pronounced acidic taste. The beer was characterized by a pH of 3.37 ± 0.01 and by the abundance of lactic acid (13.0 g/L), which was the major metabolite after ethanol (51.8 g/L), likely resulting from the catabolic conversion of wort sugars operated by LAB. Other primary catabolites observed in the beer were glycerol (3.3 g/L) and acetic acid (1.0 g/L). Microbiological counts on YPDA and WL media resulted in values of 4.39 ± 0.10 and 4.28 ± 0.3 log_10_ CFU/mL, respectively. The mMRS count was 6.00 ± 0.03 log_10_ CFU/mL confirming the primary role of LAB in the production process and their predominance over the yeast population.

### 3.2. LAB Identification and Genotyping

All LAB isolates were cocci-shaped, Gram positive, and catalase negative. They were submitted to 16S-ARDRA with three endonucleases, namely *Hha*I, *Hinf*I, and *Tru*I. These restriction enzymes were proved to be diagnostic for most LAB species frequently encountered in sour beer fermentation (Appendix A). 16S-ARDRA patterns supported the attribution of LAB isolates to *Pediococcus parvulus*/*Pediococcus damnosus*, Phylogenetic analysis of 16S rRNA gene sequences was carried out on 7 randomly selected LAB isolates. The strains formed a monophyletic group with *P. damnosus* strain DSM 20331^T^ (GenBank accession number: NR_042087.1) (Figure 2, panel A).

(GTG)_5_ rep-PCR was applied to analyze the genetic diversity within the group of LAB isolates. The patterns consisted of a minimum of 10 bands to a maximum of 13 bands with dimensions ranging from 150 to 2270 bp. The UPGMA dendrogram identified 5 singleton strains and one subcluster at 85% of rep-PCR profile similarity, suggesting high diversity of LAB strains isolated from sour beer (Figure 2, panel B).

### 3.3. Yeast Identification

A total of 50 yeast isolates were randomly selected from countable plates: 20 from YPDA and 30 from WL. Raw data of the PCR–RFLP analysis of the ITS regions are reported in Appendix A. Based on a search in the Yeast-ID database five different patterns (A to E) were attributed to the following species: pattern A to *S. cerevisiae* (100%)/*Saccharomyces cariocanus* (100%)/*Saccharomyces paradoxus*(100%); pattern B to *Pichia membranifaciens* (73%); pattern C to *Saccharomyces bayanus* (89%)/*Saccharomyces kudriavzevii* (89%)/*Saccharomyces pastorianus* (89%)/*Saccharomyces mikatae* (89%); pattern D to *Kluyveromyces blattae* (67%); and pattern E to *Brettabomyces bruxellensis* (83%).

Representative strains from each ITS PCR–RFLP pattern were submitted for sequencing of the variable domain D1/D2 at the 5′ end of the 26S rRNA gene. Specifically, strains WY104, WY117, WY203, and WY220 (pattern A) were grouped monophyletically with *S. cerevisiae* NRRL Y-12632 (97% bootstrapping) (Figure 3). Strains WY122 and WY102 (pattern B) shared the highest D1/D2 26S rRNA sequence similarity with *Pichia membranifaciens* (Figure 3). However, three substitutions (60T > C, 167C > T, and 179C > T) did not allow the monophyletic clustering of these strains with the *P. membranifaciens* strain CBS107^T^.

Strains WY115, WY205, and WY213 (pattern C) were phylogenetically related to the complex taxon *S. bayanus*, which included two well-differentiated groups of strains: *S. bayanus* var. *bayanus* (*S. eubayanus* × *S. uvarum*) and *S. bayanus* var. *uvarum* (Figure 3) [52,53]. Libkind et al. [54] proposed the use of *S. eubayanus* and *S. uvarum* as species descriptors but restricted the name *S. bayanus* to the hybrid lineages between these pure species.

Finally, strains WY59 (pattern D) and WY60 (pattern E) clustered together with *Brettanomyces anomalus* (homotypic synonym: *Dekkera anomala*) and *B. bruxellensis* and had the highest bootstrapping values (Figure 3). For strain WY59, phylogenetic results partially disagreed with the ITS PCR–RFLP analysis. This strain showed an amplified ITS product of 550 bp (Appendix A, panel A) instead of the expected 800 bp ITS product reported for *B. anomalus* in the Yeast-ID database (last accessed on 2 May 2023). Phylogenetic analysis of ITS sequences confirmed that strain WY59 formed a monophyletic cluster with *B. anomalus* CBS 77^T^ (Appendix A, panel B).

The presence of *S. bayanus*/*S. uvarum* is unusual in old-style sour beers produced at room temperature. Furthermore, hybrid strains frequently retain only one parental type of the rDNA arrays in their genome, resulting in ITS restriction profiles either of *S. cerevisiae* type or *S. eubayanus*/*S. uvarum* type [55] (Figure 4, panel A). This loss of heterozygosity in rDNA marker makes ITS-RFLP analysis unable to discriminate hybrids from their parental lineages. Based on these considerations, we performed three species-specific PCR assays, targeting *S. cerevisiae*, *S. uvarum*, and *S. eubayanus*, to confirm species attribution of strains WY115, WY205, and WY213, respectively, to *S. uvarum*/*S. bayanus* and of 35 strains of pattern A to *S. cerevisiae*. Remarkably, *S. uvarum*/*S. bayanus* sour beer strains WY115, WY205, and WY213 (pattern C) were positive in both *S. cerevisiae* and *S. bayanus*/*S uvarum* species-specific PCRs (Figure 4, panel B). Thirty-five *S. cerevisiae* isolates were positive in *S. cerevisiae*-specific PCR (Appendix A) and negative in *S. bayanus*/*S. uvarum* (Appendix A)*. S. cerevisiae* isolates, as well as the *S. uvarum*/*S. bayanus* sour beer strains WY115, WY205, and WY213 gave no PCR amplicons with an *S. eubayanus*-specific primer pair (Appendix A). Overall, the results confirmed the correct species attribution of 35 strains of pattern A to *S. cerevisiae* and that WY115, WY205, and WY213 could be *S. cerevisiae* × *S. uvarum* natural hybrids, which only retain the *S. uvarum*/*S. bayanus* rDNA copy in their genomes.

As a whole, the culturable yeast fraction of natural sour beer considered in this study consisted of *S. cerevisiae* (70%), followed by *P. membranifaciens* (16%), *S. cerevisiae* × *S. uvarum* hybrids (6%), *D. anomala* (4%), and *B.*/*D. bruxellensis* (4%) (Figure 3, panel B).

### 3.4. Yeast Genotyping

The genetic diversity of the 50 yeast isolates was estimated using the (GTG)_5_ rep-PCR method. The resulting patterns comprised 4 to 23 bands with dimensions between 200 and 2830 bp. A UPGMA dendrogram, based on Pearson’s similarity coefficient, grouped the 50 isolates into two major clusters (I and II) (Figure 5). Applying a reproducibility cutoff of 92% led to the identification of 14 biotypes, consisting of 9 subclusters (S1 to S9) and 5 singletons.

Cluster I comprised 8 yeast isolates belonging to the species *P. membranifaciens* (subclusters S1 and 2 singletons), as well as the *S. cerevisiae × S. uvarum* hybrid strain WY115 and *S. cerevisiae* strain WY220. Within cluster II, 30 *S. cerevisiae* isolates were divided into 5 subclusters (S4 to S8), while *S. cerevisiae* isolates WY112, WY111, and WY104 were singletons. *S. cerevisiae × S. uvarum* hybrids WY213 and WY205 together formed the minor cluster S3, while *B. anomalus* and *B. bruxellensis* isolates divided into two subclusters, S2 and S9.

We investigated whether the *S. cerevisiae* isolates, one belonging to each (GTG)_5_ REP–PCR subcluster, were genetically different from the commercial *S. cerevisiae* pure cultures used in primary fermentation, namely, the strains Atecrem Belgian Saison, 3724, and WLP565 (Table 1). The (GTG)_5_ rep-PCR technique did not discriminate among the commercial strains. Conversely, two alternative techniques, the inter-delta PCR assay, which amplifies LTR regions among retrotransposable elements Ty1 and Ty2, and the R3-RAPD technique, which randomly amplifies non-coding and coding DNA using the R3 sequence as a primer, showed that the *S. cerevisiae* wild isolates were genetically distinct from the commercial starters (Appendix A, panels A and B).

### 3.5. Maltose Consumption and Flocculation

A pool of 9 selected sour beer wild strains (at least one for species) were screened for their ability to consume maltose and flocculate. Maltose is the primary fermentable sugar in brewer’s wort, while flocculation can be desirable in removing yeasts after fermentation. As shown in Figure 6, *S. cerevisiae* strains WY117, WY203, and WY104 consumed maltose at the same level of glucose (*p* > 0.05), while *B. anomalus* WY59, and *B. bruxellensis* WY60 did not grow on maltose as a unique carbon source after 48 h of incubation. Unlike the conspecific strain WY102, *P. membranifaciens* WY122 was able to consume maltose (*p* > 0.05). A conventional fermentation test confirmed this result (Appendix A), which contradicted the maltose-negative phenotype described for this species [56]. Concerning the flocculation assay, no strains showed a significant flocculent phenotype (Appendix A).

### 3.6. Sporulation and MAT Genotyping

Four *S. cerevisiae* strains and 2 *S. cerevisiae* × *S. uvarum* hybrids were evaluated for their exploitability as parental lineages in hybrid construction. For this purpose, we determined sporulation efficiency and spore viability. Sporulation efficiency of *S. cerevisiae* strains varied from 22 to 23% after 3 days of incubation at 28 °C on agar acetate plates and slightly increased after 14 days (Table 2). *S. cerevisiae × S. uvarum* hybrids were not able to sporulate.

Spore viability can be defined as the ability to grow on a reach medium at 28 °C for 2 to 5 days and to produce a viable monospore colony after tetrad dissection. Even though *S. cerevisiae* strains sporulated moderately, spore viability was higher than 50%, among which *S. cerevisiae* WY203 showed the highest (Table 2).

Since ploidy status and homothallic/heterothallic lifestyle strongly affect the hybridization yield of *Saccharomyces* strains [46], we determined the *MAT* locus in each *S. cerevisiae* strain and their monospore progeny. All strains exhibited a *MAT*a/*MAT*α genotype (Table 2) and produced *MAT*a/*MAT*α monosporic progenies, suggesting that they are diploid and homothallic (*HO*/*HO*) strains (Appendix A).

### 3.7. Micro-Scale Wort Fermentation

We performed test fermentations using both maltose-consuming (*S. cerevisiae* WY104, WY117, WY203, and WY220; *P. membranifaciens* WY122; *S. cerevisiae* × *S. uvarum* WY213) and maltose non-consuming strains (*P. membranifaciens* WY102, *D. anomala* WY59, and *D. bruxellensis* WY60). To assess the kinetic variables during wort fermentation, we monitored the CO_2_ loss by weighing. *S. cerevisiae* strains exhibited the shortest lag phase and the highest fermentation efficiency values (*p* < 0.05) (Figure 7) and did not differ significantly from the *S. cerevisiae* × *S. uvarum* hybrid WY213 (*p* > 0.05). After a long lag phase, maltose-positive *P. membranifaciens* WY122 showed a fermentation efficiency and rate similar to those of *S. cerevisiae*. As expected, maltose-negative *P. membranifaciens* strain WY102 was slow to adapt to wort environment and very low in fermentation efficiency but, after a long lag phase, it exhibited high fermentative rate. *B. bruxellensis* WY60 was better than *B. anomalus* WY59 to ferment wort (*p* < 0.05).

The consumption of carbohydrates and ethanol were measured at the end of fermentation and were consistent with the fermentation curves (Figure 8). Specifically, *Saccharomyces* strains outperformed non-*Saccharomyces* ones in maltose and maltotriose usage, except for *P. membranifaciens* WY122. Similarly, *Saccharomyces* strains and *P. membranifaciens* WY122 produced higher amounts of ethanol (20.8–34.1 g/L). Notably, *S. cerevisiae* × *S. uvarum* hybrid WY213 demonstrated not only the highest ethanol production but also the highest rate of carbohydrate-to-ethanol conversion (40% *w/w*), for which the maximum theoretical value is 51%.

An HPLC analysis revealed partial consumption of dextrins, which are generally unfermentable sugars for commercial (domesticated) ale yeasts (Figure 8). *STA1* gene encodes extracellular glucoamylase which mostly contributes to dextrin consumption [50,57]. Recent studies showed that the mere presence of the *STA1* gene is not enough to be diagnostic for a “diastatic” phenotype. Krogerus et al. [50] proved that a full-length promoter assures *STA1* gene expression in truly diastatic strains, while the UAS2 region of the promoter is deleted in poorly diastatic *STA1*+ strains. Therefore, we investigated the presence of *STA1* and the related functional promoter in *S. cerevisiae* strains. As shown in Figure 9, the *STA1* gene was present in our *S. cerevisiae* sour beer strains, but the promoter-specific PCR gave a PCR product longer than that expected (630 bp instead of 599 bp). The sequencing of the promoter PCR amplicon revealed two indels and 20 substitutions in the UAS2 promoter region of *S. cerevisiae* WY117 compared to the truly diastatic strain *S. cerevisiae* 3711.

### 3.8. VOCs Determination

In the headspace of the wort and fermented samples, SPME–GC analysis revealed a total of 53 compounds detected in at least two samples (Appendix A). The VOCs occurring most frequently and abundantly were ethanol and other alcohols, esters, organic acids, aromatic compounds, furans, aliphatic hydrocarbons, and terpenoids. The PCA model revealed that two highly informative dimensions, PC1 and PC2, accounted for 61.5 and 21.1% of data variability, respectively (Figure 10, panel a). Ethanol, 3-methyl-1-butanol, and phenylethyl alcohol presented positive loadings to PC1, resulting in major contributors to the differentiation of fermented samples (Figure 10, panel b). Conversely, terpenoids like humulene and beta-myrcene contributed negatively to PC1’s distinguishing the profile of unfermented wort. A group of esters (hexanoic-, decanoic-, octanoic-, and ethanoic-acid ethyl esters) also contributed to characterizing fermented samples but with positive loadings along PC2. Fermented samples appeared spread in two groups: samples inoculated with *B. anomalus* WY59, *B. bruxellensis* WY60, and *P. membranifaciens* WY102 strains were located mainly in the second quarter of the graph at negative PC1 and positive PC2 values. This agreed with previous observations that, compared to *S. cerevisiae, Dekkera*/*Brettanomyces* spp. produces more fatty-acid ethyl esters such as octanoate and ethyl decanoate, which are responsible for fruity aroma [58]. Samples obtained by fermentation with *Saccharomyces* strains and *P. membranifaciens* WY122 were in the first and fourth quarters, at positive values for PC1. WY122 exhibited a VOC profile intermediate between *Saccharomyces* and non-*Saccharomyces* strains, characterized by the concomitant presence of a more complex ethyl ester profile and high amount of ethanol. Among the aroma-active esters, ethyl esters, which play a crucial role in determining the aroma of beer [59], were the second-most abundant VOC after acetate esters. Examples of ethyl esters were ethyl hexanoate (aroma like anise seed or apple), ethyl octanoate (sour apple), and ethyl decanoate (floral). Ethyl esters are formed intracellularly by fermenting yeast cells, and the rate of formation is highly strain-dependent and mainly affected by concentrations of the two precursors, ethanol and acyl coenzyme A, where the acid group is a medium-chain fatty acid (MCFA) [60]. During alcoholic fermentation, MCFA intermediates are prematurely released from the cytoplasmic fatty-acid synthase (FAS) complex. The key enzyme involved in the regulation of fatty-acid biosynthesis can promote the release of MCFAs, which dramatically affect the formation of ethyl esters and contribute to the differentiation of the yeast strain’s profile.

## 4. Discussion

Exploring the biodiversity of native strains found in environments related to breweries or not directly associated with them can significantly contribute to expanding the range of cultures used in craft beer production. For instance, *S. cerevisiae* strains from brewery environments have been successfully used to produce low-alcohol beer [61], but strains of the same species isolated from sourdough have also been utilized as brewing cultures to obtain the same kind of product [62]. In addition, *S. cerevisiae* has been suggested as a possible parent lineage for the construction of synthetic hybrids [46]. In this study, we applied a culturable approach to characterize the yeast fraction present in sour beer during aging in wooden barrels and proved that these indigenous yeasts have promising brewing attributes. DNA barcoding techniques and PCR fingerprinting assured the constitution of a pool of assorted yeast candidates for subsequent phenotypic assays.

Previous studies on microbial succession during the traditional lambic beer process showed that *Saccharomyces* species dominate the early stages when they sequentially consume mono-, di-, and trisaccharides, releasing yeast-associated metabolites such as ethanol, succinic acid, and methyl-1-butanol [63,64]. Depletion of maltose and maltotriose caused starvation of *Saccharomyces* species and was followed by the acidification phase [18]. In the late stage of maturation, oxidative species such as *Dekkera* spp., *B. bruxellensis*, and *P. membranifaciens* should dominate the mycobiota as they possess intra- and extracellular α-glucosidases responsible for maltooligosaccharide degradation. The sour-matured production style considered here entails inoculation with *S. cerevisiae* commercial starters to obtain a base beer that is then matured in wooden barrels. This method shortens brewing time and increases predictability compared to the traditional lambic beer process without forgoing the crucial role of the wooden barrels as an additional microbial inoculation source. To the best of our knowledge, no studies have characterized the cultivable yeast fraction from microbrewery plants based on a sour-matured production-style. Unlike the sequential succession of yeast species documented in traditional lambic beer process, in this study *S. cerevisiae* dominated the cultivable mycobiota at the maturation stage in barrels, followed by *P. membranifaciens*, *B. anomalus*, and *B. bruxellensis*. Using ITS1 metabarcoding, Shayevitz et al. [65] also demonstrated that *S. cerevisiae* is highly abundant in aged sour-ale beer samples. As *S. cerevisiae* is frequently isolated from oak wood [66], it is reasonable that this species should be a common resident in wooden barrels used for beer souring. Furthermore, *S. cerevisiae* wild strains were genetically distinct from commercial cultures used for base beer fermentation, showing that the strains are feral yeasts inhabiting the craft brewery plan. Further investigations on domestication signatures in the genome could elucidate whether they are truly wild strains or brewing strains that have evaded the domestication niche and are able to colonize natural environments [67].

The co-existence of yeast species with LAB species *P. damnosus*. *Pediococcus damnosus* is critical in sour beer production [68] as they are more hop-tolerant and low-pH-tolerant than lactobacilli [23] and are very persistent in wooden barrels [69]. *Pediococcus* spp. produce diacetyl, which is responsible for a rancid and buttery flavor. However, in craft beer this effect is attenuated by some non-*Saccharomyces* yeasts that consume diacetyl [70]. On the other hand, *Pediococcus* spp. decrease pH more slowly compared to lactobacilli, assuring that yeasts survive to complete primary fermentation [71].

During aging, the *S. cerevisiae* strains should also survive in maltose and maltotriose depletion. In co-fermentation experiments, Coehlo et al. [72] reported that both *S. cerevisiae* and *B. bruxellensis* partially consume ethanol when glucose was depleted, according to the make–accumulate–consume strategy [73]. However, *B. bruxellensis* can outperform *S. cerevisiae* in ethanol consumption, the latter being able to metabolize a wider range of carbon sources and turn to ethanol as a last resort. All our strains possess the *STA1* gene under the control of a promoter slightly longer than that found in the “diastatic” reference strain 3711. This gene encodes an extra-cellular glucoamylase (EC 3.2.1.3) which cleaves α-1,4-glycosidic bonds from the non-reducing ends of soluble maltodextrin chains releasing glucose, subsequently fermented. Deletion of the *STA1* gene reduced maltotriose consumption in *STA1*-positive beer strains belonging to lineage Beer 2 [50,57], suggesting that Sta1p contributes to maltotriose consumption in these strains. The partial degradation of dextrin detected during microscale wort fermentation supported the diastatic nature of our *S. cerevisiae* strains. Although we did not investigate *STA1* gene expression, we hypothesized that the ability to survive during beer aging can be related to a diastatic phenotype. “Diastatic” is generally considered a negative brewery trait for being responsible for over-attenuation and quality deterioration. Furthermore, diastatic strains generally have functional *PAD1* and *FDC1* genes responsible for the decarboxylation of ferulic acid into 4-vinyl guaiacol, a clove-like aroma [74]. However, in some craft beers, such as sour and Saison-style beers, 4-vinyl guaiacol and other phenolic derivative molecules, such as 4-ethyl phenol, contribute to their “funky” character by imparting spicy and barnyard flavor notes [62]. We did not specifically investigate the phenolic off-flavor (POF) phenotype, but VOC determination after wort fermentation with *S. cerevisiae* strains revealed a low amount of 4-vinyl guaiacol accounting for less than 0.4% of total VOC signals. By contrast, we found the relevant presence of phenylethyl alcohol, which contributes a fruit-like aroma. These data indicate that these strains could contribute to the formation of desirable aroma-active esters. The intermediate sporulation efficiency associated with high spore viability supported the novel *S. cerevisiae* isolates as well as their monosporic derivatives and could be exploitable in synthetic hybrid construction.

Interestingly, we isolated three strains positive to both *S. cerevisiae* and *S. uvarum*-specific PCR assays targeting housekeeping genes. Both ITS–RFLP analysis and D1/D2 26S rRNA gene sequencing indicated that these strains only retained the *S. uvarum* rDNA array. *S. uvarum* was isolated in natural European wine [75,76,77] and in cider [78,79] and South American chicha fermentations [80]. *S. bayanus*/*S. uvarum* and the hybrid species *S. pastorianus* (synonym *Saccharomyces carlsbergensis*) are extensively used to produce lager beer through bottom fermentation at low temperature [81]. Occurrence of *S. bayanus*/*S. uvarum* has been documented in the early stages of lambic beer fermentation, after which this species should disappear due to high ethanol content and low pH [20]. The results collected in this study suggested that these strains could be *S. cerevisiae × S. uvarum* hybrids. According to this hypothesis, no sporulation was observed under the conditions used in this study, suggesting that either pre-zygotic barriers or polyploidy/aneuploidy prevent meiosis in these strains. *S. cerevisiae* × *S. eubayanus* and *S. cerevisiae* × *S. kudriavzevii* interspecies hybrids have been frequently documented in craft brewing. *S. pasteurianus* was one of the most abundant species at the end of alcoholic fermentation phase in traditional lambic sour beer production [17,71] while many Trappist-style beers from Belgium are brewed with hybrids of *S. cerevisiae* × *S. kudriavzevii* [82]. This might be a consequence of winter temperatures that favor yeasts that are more cryotolerant, such as *S. kudriavzevii* and *S. pastorianus*. *S. cerevisiae* × *S. uvarum* hybrids have been documented in brewing less frequently than *S. pasteurianus* and *S. cerevisiae* × *S. kudriavzevii* interspecies hybrids. Recently, a *S. cerevisiae* × *S. uvarum* hybrid, the so-called Muri isolate, was characterized from a “kveik” culture, a traditional Norwegian farmhouse brewing yeast culture [83,84]. Dextrin consumption in microscale wort fermentation suggested that, like Muri isolate, WY213 could be *STA1*-positive.

*Pichia membranifaciens* has been identified in lambic and gueuze beers [17,71], wine [85,86,87], and other beverages [88,89]. It is one of the yeasts most frequently associated with the mycobiome of sour beer during wood aging [20,90]. The ability to form biofilm accounts for the long persistence of this species on wooden surfaces [91]. While *Pichia* spp. has been described as being involved in the biotransformation of hop terpenes such as geraniol [92], this species also produces unpleasant compounds, such as 4-ethylphenol and 4-ethylguayacol [86]. As having a non-fermentative metabolism, it is generally considered a beer contaminant and poorly attractive as a brewing culture. Here the *P. membranifaciens* strain WY122 consumed maltose and exhibited an intermediate phenotype between non-*S. cerevisiae* species such as the conspecific strain WY102, *B. anomalus*, *B. bruxellensis* and the *Saccaromyces* strains. To the best of our knowledge only a *P. membranifaciens* strain from Chinese steamed bread reportedly used maltose as carbon source [93]. Genomic characterization of strain WY122 will contribute to elucidating its taxonomic position.

## 5. Conclusions

In this study we proved that sour beer at the maturation stage can serve as reservoir of novel yeast cultures with brewing attributes. Interestingly, the *STA1*-positive *S. cerevisiae* feral strains dominated the cultivable mycobiota fraction, probably taking advantage of the ability to consume dextrins. Despite the low number of strains submitted to phenotyping, a variety of candidates—*S. cerevisiae* isolates, *S. cerevisiae* × *S. uvarum* putative hybrids, and *P. membranifaciens* WY122—were found to possess industrially desirable phenotypic traits, making them attractive candidates for both monoculture and coculture exploitation in craft beer production. Future sampling could include a higher number of microbreweries with different sour-matured beer production styles to expand the diversity of yeast strains and provide customized brewing cultures.

## Figures and Tables

**Figure 1 microorganisms-11-02138-f001:**
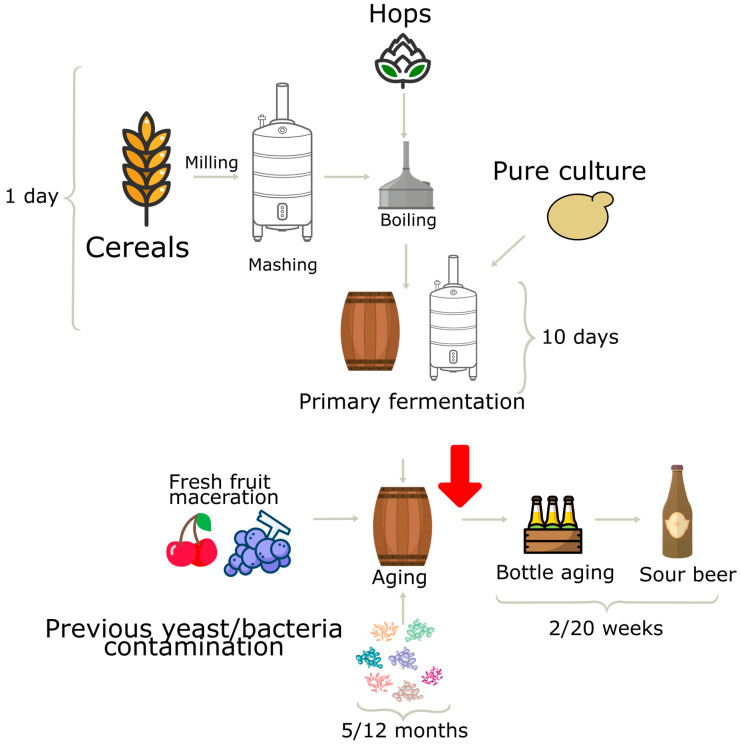
Flow-chart of semi-spontaneous craft beer fermentation. Red arrow indicates point of sampling.

**Figure 2 microorganisms-11-02138-f002:**
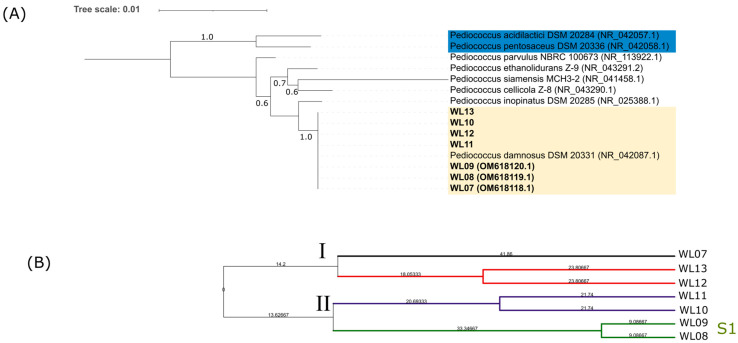
Phylogenetic and genotyping analysis of LAB isolates. (**A**) Neighbor-joining tree based on 16S rDNA sequences showing genetic relatedness between *Pediococcus damnosus* isolates and related species. The evolutionary distances were computed using the Kimura 2-parameter method and the rate variation among sites was modeled with a gamma distribution. Bootstrap values are indicated beside branches (>50%). The tree was rooted with *P. acidalictici* and *P. pentosaceous* (blue). (**B**) UPGMA clustering of LAB isolates based on (GTG)_5_ rep-PCR fingerprinting analysis. Similarities were calculated as Pearson correlation coefficient with Bionumerics software v8.10. The tree data (Newick) were exported and visualized using ITOL [38].

**Figure 3 microorganisms-11-02138-f003:**
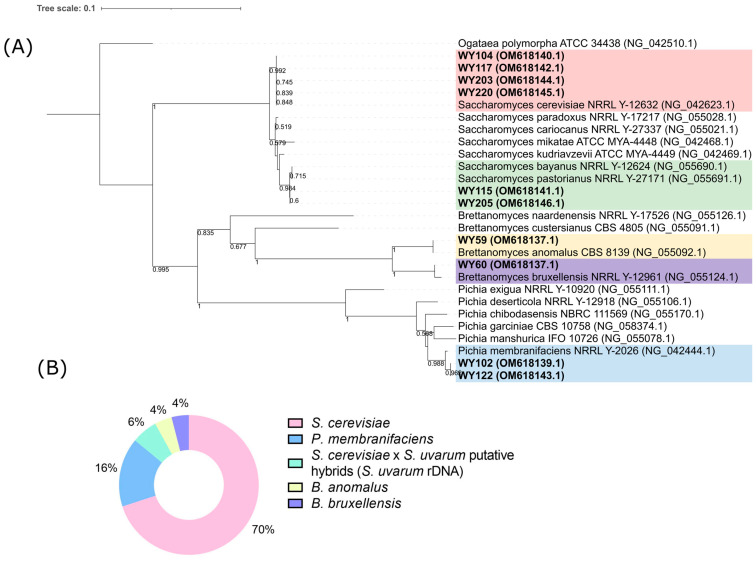
Species attribution of yeast isolates (bold). (**A**) Phylogenetic tree obtained by neighbor joining (NJ) method applied to a dataset of 28 D1/D2 26S rDNA sequences. The evolutionary distance was calculated by the Tajima–Nei method. The gamma distribution was used to model the rate of change between sites. The lengths of the branches are proportional to the number of nucleotide substitutions, and they have been measured using the divergence scale shown at the top left. Bootstrap values (1000 replicates) are indicated beside branches (>50%). The tree was rooted with *Ogataea polymorpha* as outgroup. The tree data (Newick) were exported and visualized using ITOL [38]. Colors of backgrounds were as follows: pink, *S. cerevisiae*; blue, *P. membranifaciens*; green, *S. cerevisiae* × *S. uvarum* hybrids; yellow, *D. anomala*; purple, *D. bruxellensis.* (**B**) Pie-chart representing species abundance in sour beer: pink, *S. cerevisiae*; blue, *P. membranifaciens*; green, *S. cerevisiae* × *S. uvarum* hybrids; yellow, *D. anomala*; purple, *D. bruxellensis*.

**Figure 4 microorganisms-11-02138-f004:**
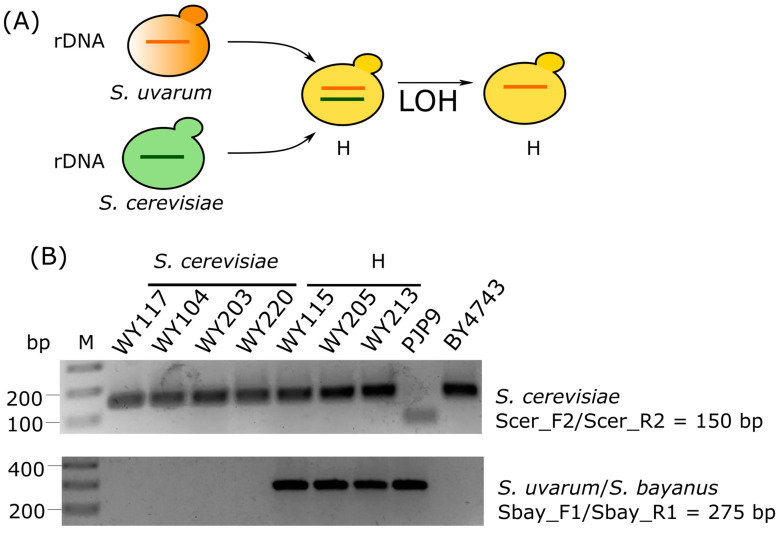
Characterization of *Saccharomyces* strains by species-specific PCR assays. (**A**) Illustration of loos of heterozygosity (LOH) occurring in rDNA arrays of hybrid lineages. (**B**) *S. cerevisiae*-specific and *S. bayanus/S. uvarum*-specific PCR assays with Scer_F2/Scer_R2 primers targeting *MEX67* gene (amplicon size 150 bp) and primers Sbay_F1/Sbay_R1 targeting *DBP6* gene (amplicon size 275 bp), respectively. All tested strains were negative in PCRs targeting *S. eubayanus*-specific *FSY1* gene (amplicon size 228 bp) (Appendix A). *S. cerevisiae* BY4743 and *S. uvarum* PJP9 were used as internal control. *S. cerevisiae* strains WY117, WY104, WY203, and WY220 were selected as representative of isolates with pattern A. Abbreviations: M, molecular weight marker; S. cer, *S. cerevisiae*; Sbay, *S. uvarum*/*S. bayanus*; H, hybrid; LOH, loss of heterozygosity.

**Figure 5 microorganisms-11-02138-f005:**
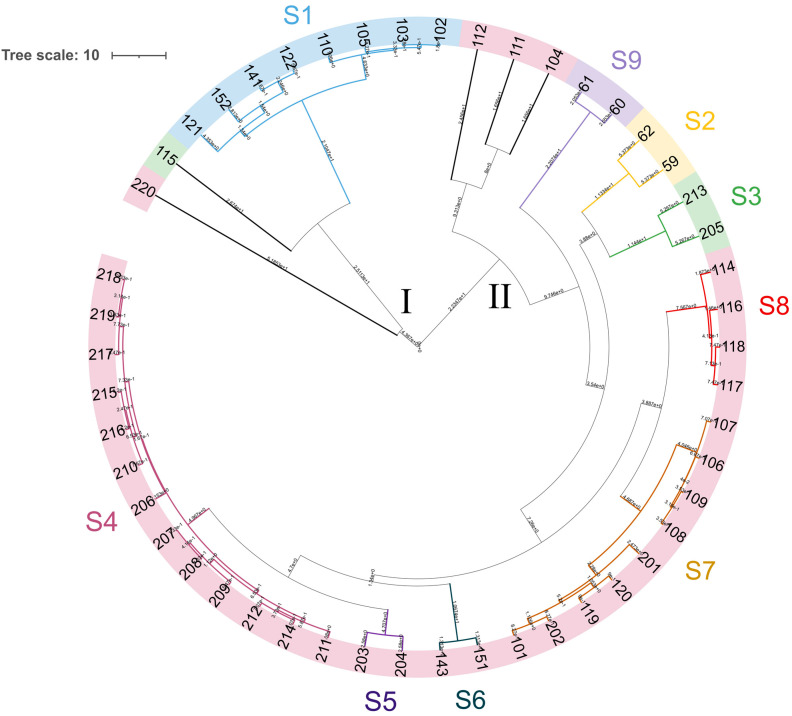
Dendrogram generated from (GTG)_5_ rep-PCR fingerprints of 50 sour beer wild yeasts. Similarity percentages were calculated using Pearson correlation coefficient, while hierarchical clustering analysis was carried out using the UPGMA (unweighted pair-group method with arithmetic mean) method with Bionumerics software v8.10. The tree data (Newick) were exported and visualized using ITOL [38]. The similarity value above 92% was used to discriminate biotypes numbered from S1 to S9 and singletons. The color codes are attributed as follows: pink, *S. cerevisiae*; blue, *P. membranifaciens*; green, *S. cerevisiae* × *S. uvarum* hybrids; yellow, *D. anomala*; purple, *D. bruxellensis*.

**Figure 6 microorganisms-11-02138-f006:**
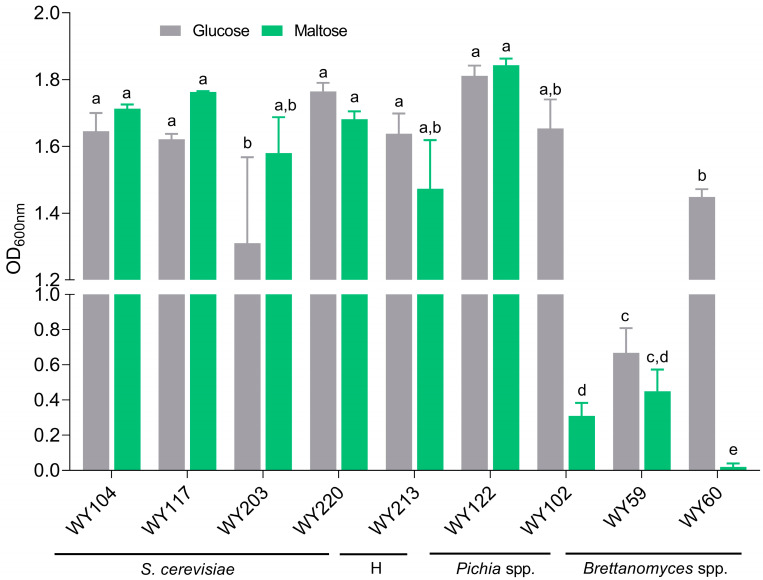
Maltose (green) and glucose (grey) consumption tests. Statistical differences are indicated with different letters (two-way ANOVA; *p* < 0.05). Abbreviation: H, *S. cerevisiae* × *S. uvarum* hybrid.

**Figure 7 microorganisms-11-02138-f007:**
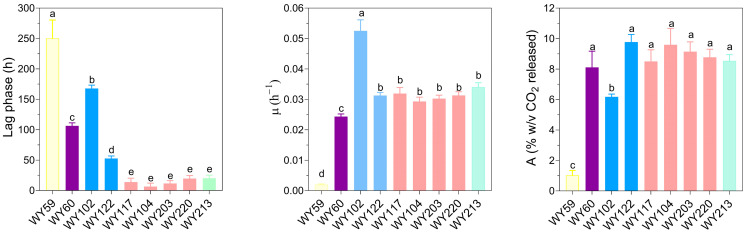
Kinetic parameters (lag phase λ, fermentation rate μ, and maximum fermentation efficiency) of yeast candidates in micro-scale wort fermentation (11.5 Plato; 20 °C). Statistical differences are indicated with different letters (one-way ANOVA; *p* < 0.05). Species was represented as follows: pink, *S. cerevisiae*; blue, *P. membranifaciens*; green, *S. cerevisiae* × *S. uvarum* hybrids; yellow, *B. anomalus*; purple, *B. bruxellensis*.

**Figure 8 microorganisms-11-02138-f008:**
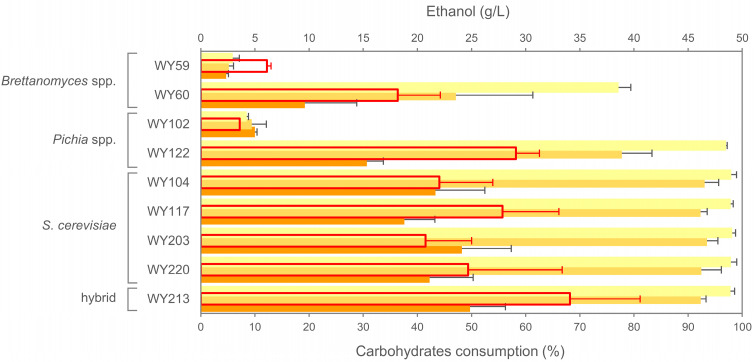
HPLC analysis of microscale fermentation broths inoculated with selected yeast strains. % consumption of maltose (yellow), maltotriose (light orange) and maltodextrins (orange); ethanol formation (g/L; red). Hybrid: *S. cerevisiae* × *S. uvarum*.

**Figure 9 microorganisms-11-02138-f009:**
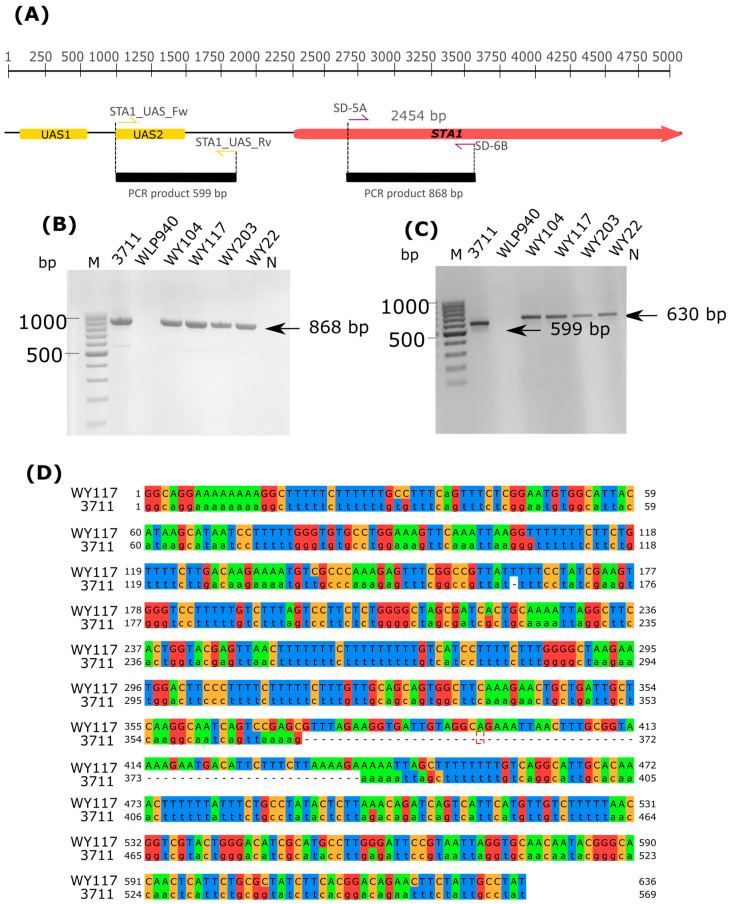
Screening for the *STA1* gene and related promoter in *S. cerevisiae* sour beer strains. (**A**) Strategy to PCR amplify *STA1* open reading frame and the promoter region with UAS_1 and UAS_2 located upstream the activation sequence [50]. (**B**) Individual PCR reactions using primers SD-5A/SD-6B with 3711 as reference strain with “diastatic” phenotype (*STA1*+, no deletion in *STA1* promoter) and WLP940 (*STA1*−) as reference strain with no diastatic phenotype. (**C**) Individual PCR reactions using primers STA1_UAS_Fw/STA1_UAS_Rv with 3711 as diastatic reference strain (*STA1*+, no deletion in *STA1* promoter) and WLP940 reference strain with no “diastatic” phenotype *(STA1*−) as no diastatic control. (**D**) Nucleotide sequence alignment produced by software Jalview v2.11.2.0 showing UAS2-2 region of WY117 and 3711. Promoter region of diastatic control strain 3711 was retrieved from [50].

**Figure 10 microorganisms-11-02138-f010:**
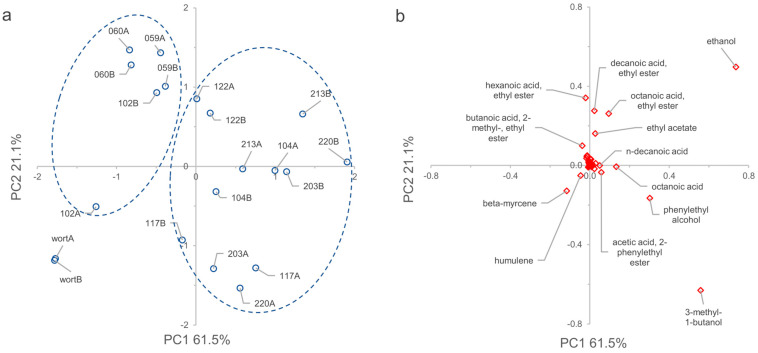
VOC profiles of wort and micro-fermented samples reconstructed by headspace SPME–GC–MS analysis, followed by principal component analysis (PCA). (**a**) Scores plot (PC1 vs. PC2) accounting for 61.5 and 21.1% of variability, respectively. (**b**) Loading plot of the volatile molecules. The compounds with the highest loading value in PC1 and PC2 are labeled. A and B are replicated samples.

**Table 1 microorganisms-11-02138-t001:** Reference strains used in this work.

Strain	Species	Features	Source
BY4742	*S. cerevisiae*	*MAT*α; *his3*Δ1; *leu2*Δ0; *lys2*Δ0; *ura3*Δ0	Euroscarf
BY4743	*S. cerevisiae*	*MAT*a/α; *his3*Δ1/*his3*Δ1; *leu2*Δ0/*leu2*Δ0; *met15*Δ0/*MET15*; *LYS2*/*lys2*Δ0; *ura3*Δ0/*ura3*Δ0	Euroscarf
3711	*S. cerevisiae* var. *diastaticus*	*STA1* positive French Saison ale yeast	Wyeast
WLP940	*S. pastorianus*	*STA1* negative lager yeast	White Lab
Atecrem	*S. cerevisiae*	Atecrem Belgian Saison	Bioenologia 2.0
WLP565	*S. cerevisiae*	Belgian Saison I Ale Yeast	White Lab
3724	*S. cerevisiae* var. *diastaticus*	Belgian Saison Yeast	Wyeast
PJP9	*S. uvarum/S. bayanus*	nd	Gift from P. Marullo

Abbreviation: nd, not determined.

**Table 2 microorganisms-11-02138-t002:** Sporulation efficiency and spore viability of *Saccharomyces* wild sour beer strains. Abbreviation: nd, not determined.

Species	Strains	Sporulation Efficiency (%)		Spore Viability (%)	*MAT* Loci
		3 d	14 d		
*S. cerevisiae*	WY104	22.96	24.56	75	*MAT*a/*MAT*α
	WY117	23.84	32.72	75	*MAT*a/*MAT*α
	WY203	22.70	27.85	81	*MAT*a/*MAT*α
	WY220	22.78	32.13	56	*MAT*a/*MAT*α
hybrids	WY205	1.22	0.89	nd	nd
	WY213	2.48	0.41	nd	nd

## Data Availability

The sequence data generated in this study have been submitted to NCBI database under accession numbers OM618137 to OM618146 and OM618118 to OM618120.

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
