# Peer review of "Sour Beer as Bioreservoir of Novel Craft Ale Yeast Cultures"

_microorganisms, 2023, doi:10.3390/microorganisms11092138_

Round 1

Reviewer 1 Report

The study entitled “Sour beer as bioreservoir of novel craft ale yeast cultures” is about novel ale yeast cultures for craft brewing. Sour beer aged in wooden barrel is selected as source of novel ale yeasts. The material and methods of this study is well planned and explained. The different approaches used to characterize the yeasts, from genetic, metabolomic and phylogenetic point of view make the study of good scientific relevance. I have just few comments and conclusions must be improved. However, I think that findings from this study are hardly reproducible, due to different variables influencing the microbiota of sour beer. Anyway, I consider this study suitable for the publication after minor revisions.

Line 37à off flavours generally refer to negative compounds. Therefore, I’m slightly doubtful about this statement. I don’t think that off-flavours may increase the perceived quality od craft beer. Please, rephrase or better explain the meaning of the sentence

Line 56à please, report the references only as numbers

Figure 1à barley or barley malt is it use as raw materials? If barley malt, please change the text in the figure

Conclusionsà any limitations of the study and future perspectives? In my opinion, conclusion seems too short and incomplete.  

Reviewer 2 Report

The manuscript presented by Nasuti et al., describes the identification procedure and phenotypic characterisation of yeast isolated from sour beer. Despite the manuscript not having scientific novelty, it is a thorough piece of work that deserves publication if revised. Please see below my main and specific comments.

Main comments:

The authors need to describe throughout the manuscript the specific source of the isolates investigated. They mention in the methods the types of beer, but they don’t specify from which beers where the specific isolates obtained from (beer 1, 2 and 3?). The authors also need to justify in each section why are they picking a specific sub-set of isolates for phenotypic characterisation. There is also a lot of discussion included in the results section that needs to be either removed or both sections combined as one. As an example, lines 568-576 are discussion.

Specific comments:

Line 47: Fix citation, X et al.

Line 53: OGM? You mean GMO?

Table 1: Strain WLP940 species is Lager?

Line 293: Presenting a size? Fix sentence and citation.

Line 310: Predominance over, not on.

Line 312-318: Where is this data? Please include supporting data.

Line 319: 7 isolates selected but Figure 2A only shows 3. Please include phylogenetic analysis for the 7 isolates.

Line 319: Randomly selected, but from which samples?

Line 344: Randomly selected, but from which samples?

Line 354: Isolate WY115 is Pattern C in table S3. Please clarify.

Line 360: Fix sentence, grammatically incorrect.

Line 388: Full stop missing.

Line 404: Data for S. cerevisiae strain BY4742 is not presented in Figure 4b. Authors mention 35 S. cerevisiae strains, the data is not presented in the figure. Please include results for strain BY4742 and the rest of the isolates.

Line 408-410: Data for eubayanus specific PCR is not presented in figure 4B. Please include negative PCR results with a positive control. 

Figure 4: There is a band for strain PJP9. Please explain in text.

Line 458: Why those 9 isolates? Justify in text.

Line 549: Fix 3-methyl-1-butanol to match with Fig 10 (1-butanol 3-methyl)

Line 583-587: This paragraph needs rewriting, its not readable. 

Line 612-614: This is too speculative. Yes, they are both from wood from similar trees but the environment is completely different. 

Line 640-644: Isnt 2-methoxy-4-vinylphenol the same as 4-vinyl guaiacol? Figure S5 shows high production of 2-methoxy-4-vinylphenol. The authors need to clarify this, and if this is the case then rewrite the abstract, results and discussion to reflect this change in interpretation of the data.

Fix as specified in the reviews.

Round 2

Reviewer 2 Report

Edits are sufficient for publication.